# Optimization of Short-Term Hot-Water Treatment of Apples for Fruit Salad Production by Non-Invasive Chlorophyll-Fluorescence Imaging

**DOI:** 10.3390/foods9060820

**Published:** 2020-06-22

**Authors:** Werner B. Herppich, Marco Maggioni, Susanne Huyskens-Keil, Tina Kabelitz, Karin Hassenberg

**Affiliations:** 1Department of Horticultural Engineering, Leibniz Institute for Agricultural Engineering and Bioeconomy (ATB), Max-Eyth-Allee 100, 14469 Potsdam, Germany; maggionsss@gmail.com (M.M.); tkabelitz@atb-potsdam.de (T.K.); khassenberg@atb-potsdam.de (K.H.); 2Division Urban Plant Ecophysiology, Research Group Quality Dynamics/Postharvest Physiology, Humboldt-Universität zu Berlin, Lentzeallee 55/57, 14195 Berlin, Germany; susanne.huyskens@hu-berlin.de; 3Department of Agricultural and Food Sciences, Alma Mater Studiorum—University of Bologna, Viale Giuseppe Fanin 40, 40127 Bologna, Italy

**Keywords:** fruit quality, postharvest physiology, processing, pigments, heat stress

## Abstract

For fresh-cut salad production, hot-water treatment (HWT) needs optimization in terms of temperature and duration to guarantee a gentle and non-stressing processing to fully retain product quality besides an effective sanitation. One major initial target of heat treatment is photosynthesis, making it a suitable and sensitive marker for HWT effects. Chlorophyll fluorescence imaging (CFI) is a rapid and non-invasive tool to evaluate respective plant responses. Following practical applications in fruit salad production, apples of colored and of green-ripe cultivars (‘Braeburn’, ‘Fuji’, ‘Greenstar’, ‘Granny Smith’), obtained from a local fruit salad producer, were hot-water treated from 44 to 70 °C for 30 to 300 s. One day after HWT and after 7 days of storage at 4 °C, CFI and remission spectroscopy were applied to evaluating temperature effects on photosynthetic activity, on contents of fruit pigments (chlorophylls, anthocyanins), and on various relevant quality parameters of intact apples. In ‘Braeburn’ apples, short-term HWT at 55 °C for 30 to 120 s avoided any heat injuries and quality losses. The samples of the other three cultivars turned out to be less sensitive and may be short-term heat-treated at temperatures of up to 60 °C for the same time. CFI proved to be a rapid, sensitive, and effective tool for process optimization of apples, closely reflecting the cultivar- or batch-specificity of heat effects on produce photosynthesis.

## 1. Introduction

The fact that fruit skin remains on apple slices prepared for fruit salads [1] increases the risk of microbiological contamination because microbes, including human pathogens, are naturally located on product surfaces [2]. For long-term storage, hot-water treatment (HWT) is known as a rapid, inexpensive, effective, and gentle sanitation technique avoiding any additional chemicals [3,4]. Furthermore, HWT may even improve maintenance of fruit quality in storage [5,6,7,8] and is suitable for both conventional and organic products [9,10].

This technique has also been recommended for fresh-cut fruit salad production; here, however, HWT needs adaptation [11]. Treatment temperature and duration, in particular, must be optimized to efficiently but sustainably inactivate microbial spoilage organisms and human pathogens, but not to adversely affect produce metabolism and, thus, internal quality [8]. Major initial target of heat in apple skin cells is the photosynthetic apparatus [12,13]. Consequently, photosynthesis is a suitable and sensitive marker for heat-stress effects [14,15]. In general, chlorophyll fluorescence imaging (CFI) and spectral analyses provide rapid and non-invasive means for effective process optimization [12,13,16,17]. In this context, it was noted [18] that the effects of heat-treatment on photosynthetic performance, especially the heat-impacts on photosystem II (PSII), was well investigated, while the potential “reversibility of heat-driven PSII inhibition” has only infrequently been studied.

Investigations that comprehensively evaluate fruit-specific thresholds for the optimal non-destructive treatment conditions of short-term hot-water treatments (sHWT) and the potential shelf-life temperature-stress responses of apples are still lacking [11]. In addition, reactions of apples to HWT may be cultivar-specific [19,20,21,22], which was shown to be important for choosing the optimal processing conditions [21].

Thus, fruit of two colored (‘Braeburn’ and ‘Fuji’) and two green-ripe apple cultivars (‘Greenstar’ and ‘Granny Smith’), practically relevant for fruit salad production, were short-term hot-water treated in the range of 44 to 70 °C for 30 to 300 s. Afterwards and after additional 7 d of storage at 4 °C, intact apples were analyzed for changes in CFI parameters and important spectral factors (normalized difference vegetation index, NDVI; normalized anthocyanin index, NAI) to evaluate the potential effects of combinations of processing temperature and duration on the intactness of the photosynthetic apparatus, the contents of essential apple fruit pigments (chlorophylls and anthocyanins) and various important parameters of fruit quality (tissue elasticity and strength, and vitamin C, soluble solids, and titratable acid contents). Results will, in particular, be evaluated in terms of physiological heat-stress responses, potential practical implications and the cultivar specifity of the reactions.

## 2. Materials and Methods

### 2.1. Plant Material and Short-Term Hot-Water Treatment

Apples (*Malus domestica* Borkh.) of the “colored” cultivars ‘Braeburn’, ‘Fuji’, and the “green-ripe” cultivars ‘Greenstar’ and ‘Granny Smith’, practically used for commercial fresh-cut fruit salad, were directly obtained from a producer (mirontell fein & frisch AG, Großbeeren, Germany) to work as close to practice as possible. For the experiments, only apples free of any defects were selected for similar size and color.

Initially, relevant quality parameters (chlorophyll fluorescence (CFA) and spectral (PA) parameters, tissue elasticity, and strength, vitamin C (vit C), total soluble solids (TSS), and titratable acidity (TA) contents, see below) were analyzed on nine untreated apples. Then, 6 apples per group (i.e., 51 apples per cultivar) were short-term hot-water-treated (sHWT) in the temperature range of 44 to 64 °C for 30 to 300 s, while 6 untreated samples were used as controls.

HWT were performed as previously described [2,11] in detail. Apples were heat-treated exactly at the distinct temperatures (experiment 1: 46 to 64 °C at 2 K intervals, exp. 2: 50 to 60 °C at 1 K intervals; exp. 3–7: 42, 47, 51, 53, 55, 57, 60, and 65 °C; exp. 8: 42, 47, 51, 53, 55, 57, 60, 65, and 70 °C; exp. 9: 55 °C) and durations (30, 60, 120, 180, and 300 s) within a stainless steel container (GN container, ½-150, BLANCO professional GmbH, Oberderdingen, Germany) fixed in a commercial water bath (GFL 1086, Gesellschaft für Labortechnik mbH, Burgwedel, Germany), both filled with tap water. The additional GN container simplified water replacement and helped to avoid cross-contamination between fruit. Temperatures in the water bath and in the container were controlled with thermometers and reset before the start of the next treatment. During HWTs, apples were fully immersed in water with help of a stainless steel plate. Details on heat distribution dynamics during treatments are given in [2].

After treatments, apples were stored at 4 °C overnight before non-destructive analyses (CFA, PA) were performed on three apples randomly chosen per group. These apples were then subjected to destructive analyses the next day. The other samples were stored at 4 °C for 7 d to allow for potential recovery effects. All measurements were repeated after 8 d (non-destructive) and 9 d (destructive), respectively.

### 2.2. Spectral Analyses

With a hand-held spectrophotometer (Pigment Analyzer PA110, CP-Control in Applied Physiology GbR., Dallgow-Döberitz, Germany) at three positions in equatorial region (distance 120°) of the apples, remission spectra in the wave length range of 400 to 1100 nm were recorded and two relevant indices were evaluated. The Normalized Difference Vegetation Index (NDVI) calculated as (R780 − R660)/(R780 + R660) is a close indicator of the chlorophyll content [23,24], while the Normalized Anthocyanin Index (NAI) = (R780 − R570)/(R780 − R570) reflects variations in the anthocyanin content of the fruit [25,26].

### 2.3. Chlorophyll Fluorescence Imaging

With the open chlorophyll fluorescence imaging system FluorCam FC 800-O (PSI—Photon Systems Instruments, Drasov, Czech Republic), images of the basic (F_0_) and the maximum (F_m_) fluorescence signals of dark adapted (ca. 15 min) fruit were monitored and from these the variable fluorescence (F_v_ = F_m_ − F_0_) calculated pixel-wise. The ratio F_v_/F_m_ reflects the potential maximum photochemical efficiency or photon yield of photosystem II (PSII) and, thus, indicates the potential photosynthetic capacity and the intactness of the photosynthetic machinery [15].

### 2.4. Internal Quality Parameters

Elasticity and strength of fruit skin and tissue were evaluated with a TA. XT plus texture analyzer (Stable Micro System, Godalming, UK). From force-deformation curves (pre and post-penetration speed, 20 mm s^−1^; penetration speed, 4 mm s^−1^; fixed maximum penetration: 11 mm; T, ca. 20 °C), three parameters were extracted: 1) the maximum force (F_max_) representing the strength of the skin, 2) the average force (F_av_) representing the strength of the flesh, and 3) the slope representing the elastic response of the tissue often called “stiffness” [27].

For vitamin C measurements, the specific RQflex refractometer test stripe was dipped into the apple juice and then the reflection measured by RQflex® 2 Reflectoquant® reflectometer (Merck KGaA, Darmstadt, Germany). Values of ascorbic acid are given in [g m^−3^]. A DR301-95 refractometer (A.Krüss Optronic, Hamburg, Germany) was used to measure the total soluble solids (TSS) content of the fruit juice. Total soluble solids contents are expressed in %.

Titratable acidity (TA) was measured by titration (T50 titrator, Mettler Toledo; Gießen, Germany) of 0.5 mL juice with 0.01 mol L^−1^ NaOH solution to pH 8.2 and expressed as malic acid equivalents (kg m^−3^).

### 2.5. Statistical Analysis

All data were statistically analyzed (ANOVA) with WinSTAT (v2018; R. Fitch Software, Staufen, Germany). Significance of differences between means was evaluated by Duncan’s multiple range test (*p* < 0.05).

## 3. Results

### 3.1. Effects of Heat Treatments on Photosynthetic Activity of Apples

In apples of all four cultivars, the potential maximum photochemical efficiency of PSII, F_v_/F_m_, started to significantly decline when the HWT temperature was increased above a certain threshold, even at the short-term duration of only 30 s (Figure 1). After sHWT above 60 °C, effects were irreversible and photosynthesis of the apples did not recover during storage at 4 °C for 7 d.

Effects of HWT on the intactness of the photosynthetic apparatus seemed cultivar-specific. For ‘Fuji’, ‘Granny Smith’, and ‘Greenstar’ apples, only temperatures above 60 °C proved to be critical for a functional photosynthesis under the chosen conditions. However, in samples of ‘Braeburn’, F_v_/F_m_ already significantly declined at a threshold of 55 °C (Figure 1). Closer inspection further indicated (data not shown) that one week of storage at 4 °C led to minor positive effects, i.e., partial recover of F_v_/F_m_, in apples of this cultivar when treated at temperatures between the critical 55 and 60 °C.

Because photosynthesis of ‘Braeburn’ apples turned out to be most sensitive to HWT, photosynthetic heat responses were analyzed in more detail in fruit of this cultivar (Figure 2). In these experiments, treatments for 60 s did not further affect the overall temperature response of F_v/_F_m_ as compared to those shown in Figure 1 (i.e., 30 s-treatments). Again, potential maximum photochemical efficiency only declined when treatment temperature was increased above 55 °C. HWT for 60 s, nevertheless, resulted in somewhat lower photosynthetic performance at the highest temperatures, but differences were statistically not significant (c.f. Figure 1, Figure 2a).

The observed changes in F_v/_F_m_ were due to the variations of both F_v_ (Figure 2b) and F_m_ (Figure 2c), but not F_0_ (Figure 2d). The evaluation of these respective parameters provided further insight into the potential physiological background of the photosynthetic variations; however, it did not improve the overall analysis of the physiological responses to heat. In contrast, it introduced additional variability (c.f. Figure 2a vs. Figure 2b). Thus, the further use of F_v_/F_m_ for this purpose is sufficient and meaningful.

While HWT at 30 and 60 s provoked very similar responses, extending the treatment duration at the critical temperature of 55 °C beyond 60 s resulted in a significant and pronounced decline of F_v_/F_m_, i.e., in the progressive reduction of the photosynthetic competence of the fruit (Figure 3). At a very high temperature of 70 °C, even HWT of only 30 s led to a nearly complete breakdown of the maximum photochemical efficiency (Figure 3).

In addition, heat responses occurred equally over the entire fruit, as shown in Figure 4. Here, pixel-averaged means of F_v_/F_m_, analyzed separately for each of the two sides of three ‘Braeburn’ apples per HW-treatment are presented. For the measurements, the two sides were as closely as possible identified as red or yellowish-green, i.e., with more or less complete flush colors, respectively. However, the stage of color development in the mature ‘Braeburn’ apples did neither affect the heat-stress response of the fruit nor did it hamper the measurement with the applied technique. This is further highlighted by the images of F_v_/F_m_ as presented in Figure 5.

This all indicates that the means of the fluorescence parameters truly reflected the actual total variation of the photosynthetic activity and not a locally restricted temperature response (c.f. e.g., results of F_v_/F_m_ at HWT 65 °C). This was also valid for all fruit after one week of storage (Figure 5). Thus, averaging over the entire fruit yielded valid results also in the case of colored apples. In addition, it also proved that the HW-treatment technique used indeed allows for uniform heating of the complete fruit surface.

### 3.2. Effects of Heat Treatments on Relevant Apple Pigments

Analyses of the Normalized Difference Vegetation Index (NDVI) (Figure 6) indicated pronounced differences in chlorophyll contents between fruit of the colored and the green-ripe cultivars. In particular, in the deep-green ‘Granny Smith’ apples, NDVI was high and standard deviation low, while in samples of the yellowish-green ‘Greenstar’, the NDVI was much closer to that of fruit of the colored cultivars. Nevertheless, irrespective of the cultivar investigated, no effects of HWT on NDVI and, thus, chlorophyll content were evident. Chlorophyll content was also not affected by storage and NDVI remained constant; therefore, results of NDVI for 8 days after treatments were not shown.

Similarly, results of Normalized Anthocyanin Index (NAI) were distinctly different in fruit of the green-ripe and the colored cultivars being higher in ‘Braeburn’ and ‘Fuji’ apples with their clearly high and variable anthocyanin contents (Figure 7). In samples of ‘Granny Smith’, NAI was also relatively high and, again, showed only a low variability. However, in neither cultivar any effect of HWT on NAI and thus anthocyanin content was measured. In general, if there were any cultivar-specific treatment effects on photochemical efficiency of photosystem II, they occurred to be independent of color or pigment contents of fruit.

### 3.3. Effects of Heat Treatments on Relevant Quality Parameters of Apples

The statistical evaluation of results of vitamin C, total soluble solids, and titratable acidity contents, and tissue strength and stiffness did not reveal any effect of HWT on any of those parameters. This was valid for all measurements and all cultivars. Therefore, the respective data were pooled (Table 1).

Although there were significant differences between the samples of the different cultivars, only in ‘Braeburn’ apples, the vitamin C content declined during cold-storage (4 °C). In addition, TSS slightly varied without a clear trend in samples of the other cultivars. Titratable acidity, given as malic acid equivalents (kg m^−3^) did not vary at all during storage. Similar to vitamin C content, the mechanical properties, tissue strength, and stiffness declined during cold storage (4 °C) only in ‘Braeburn’ apples.

## 4. Discussion

### 4.1. Temperature Effects on Fruit Quality

Short-term (up to 60 s) hot-water treatments (sHWT) of up to 65 °C did not negatively affect the quality of fruit of all four apple cultivars. This includes important parameters such as vitamin C, soluble solids and titratable acidity contents, but also biomechanical properties and pigments relevant for peel colors. Thus, this treatment obviously did not influence taste, texture, and appearance of the treated apples, irrespective of the cultivar. This finding confirms results reported on apples and other fruits e.g., [28,29,30,31]. On the other hand, this finding can easily be explained by the fact that sHWT as used here could not heat up the fruit body but only the outer epidermal and subepidermal tissues. In fact, most of the tissue of the apple fruit could not be warmed at all due to the short duration of the treatments as recently shown by heat transfer analyses [2,32]. As the above quality parameters were measured either on fruit juice obtained from fruit pulp tissue or on the entire fruit, effects of sHWT were, thus, not highly probable.

Nevertheless, the epidermal and some subepidermal tissue cells, only representative for tissue few millimeters into the fruit body, may have experienced some heat stress [2]. This small volume is most important for photosynthesis of apples [26,33]. Spectroscopic evaluation of chlorophyll and anthocyanin contents by means of NDVI [24] and NAI [25,26] analyses, however, did not reveal any temperature effects on the contents of these relevant pigments, mostly concentrated in the cells of the peel tissue [26,33,34,35]. These results demonstrate that the short time the fruit were exposed to heat was not enough to induce mechanisms leading to the degradation of these pigments. This is in agreement with previous studies investigating the potential temperature-induced changes in peel color and indicated that mild heat treatment did obviously not negatively affect epidermal pigments [32,36], but instead, drastically increased pigment contents in heat-stressed leaves of *Zea mays* [37].

At least in the yellow-red ‘Braeburn’ apples, some changes in mechanical properties were observed. However, these variations in both strength and stiffness occurred virtually irrespective of the heat treatment. Hence, they are more probably related to advanced fruit development than to processing. Although this seems speculative, progressed maturation (i.e., unripe, ripe, and overripe) of fruit of this colored cultivar at harvest could also explain their higher stress sensitivity as was previous shown for long-term heat treatments for red ‘Cortland’ and green-red ‘Jonagold’ apples [38], but not in unripe developing fruit of other cultivars heat-stressed under irradiation (i.e., presumably light-stressed) at 500 µmol m^−2^ s^−1^ [39]. In addition, ‘Braeburn’ apples are well-known for rapid reduction of texture with fruit development and during shelf-life [40,41].

### 4.2. Are sHWT Effects Cultivar-Specific?

In contrast to the above quality parameters, analyses of CFI rapidly and sensitively demonstrated that sHWT indeed impacted the primary physiological activity of apples, indicated by the evaluation of their overall photosynthetic performance, when fruit were heated beyond 55 or 60 °C, respectively. The thresholds of heat inhibition, however, varied between samples of the four cultivars in the presented study. In this context, the threshold observed in ‘Braeburn’ apples was distinctly lower than those of ‘Fuji’, ‘Granny Smith’, or ‘Greenstar’. This may point out a pronounced difference in the direct physiological responses of apples of different cultivars to heat treatments. Similar findings on the cultivar-related variability of HWT-responses of apples were also reported by other authors (e.g., [19,20,21,38]).

Comparing the effects of long-term (15 min to 4 h) hot-water treatments (40, 45, 51 °C) on e.g., browning, firmness, respiration, TSS, and TA of eleven apple cultivars, Kim et al. [19] found marked differences in the cultivar-specific responses of samples to duration and temperature. While apples of ‘Golden Delicious’ and ‘Delicious’ were relative tolerant to the heat treatments, those of ‘Liberty’, ‘Monroe’, ‘RI Greening’, and ‘Rome’ were highly susceptible. Nevertheless, the authors could not identify a possible explanation for the diversity in responses.

In addition, measuring browning, firmness, titratable acidity, TSS, ethanol production, and chlorophyll fluorescence (CF), Fan et al. [38] found that ‘Jonagold’ apples were less sensitive to hot air treatments (46 °C, exposed for 0, 4, 8, 12 h, then stored for 3 months) than those of ‘Cortland’. In addition, they showed that fruit harvested overripe were generally more sensitive than ripe or immature samples. Although not analyzed further, the authors stated that the stage of development may directly influence the sensitivity of fruit to heat treatments.

Visible analyses of the effects of HWT-rinsing (52−58 °C for 15 to 120 s) on apples of four cultivars after CA storage for 6 months clearly indicated a significantly higher heat resistance of ‘Gala’ and ‘Topaz’ than of ‘Golden Delicious’ or ‘Pinova’ fruit, irrespective of temperature and duration of the treatment [21].

In the present study on apples practical relevant for fruit salad production, fruit of the two green-ripe cultivars seemed to be less sensitive to sHWT than red-colored samples. This might point to some influence of fruit (peel) color on the sensitivity to HWT effects. Indeed, there is evidence that long-term heat treatments are effective in retaining fruit quality in red apples but less so in fruit of green cultivars [7,30]. However, results of other studies, summarized above, showed that the fruit color of the respective cultivars, i.e., whether apples were green-ripe or red-colored, does not affect fruit responsiveness to heat.

Nevertheless, it seems clear that the stage of fruit development is an important determinant of heat stress sensitivity [38]. In this context, the fact that heat stress immediately and rapidly leads to a large respiratory burst (e.g., [18]) that needed to be fed by stored carbon substrates may be interesting. It may be speculated that the pool of compounds available for respiration could be relevant. However, except ‘Fuji’, apples of the other cultivars showed nearly the same mean TSS. Although samples of ‘Granny Smith’ always had the highest content on malic acid equivalents, this parameter was very similar in fruit of the other three cultivars. Consequently, it is very improbable that carbon reserves may be a relevant factor determining sensitivity to heat treatments. Similarly, results presented by Shao et al. [20] could also not reveal any influence of either TSS or TA on the effects of long-term heat treatment (38 °C for 4 d) on storage or shelf-life quality ‘Gala’, ‘Golden Delicious,’ and ‘Red Fuji’ apples. In addition, in the study of Fan et al. [38], sensitive overripe apples contained less respirational resources than unripe or ripe-harvested samples. This, however, was valid both for “heat-stress sensitive” ‘Cortland’ and for “resistant” ‘Jonagold’ apples.

Thus, it is probable that both long-term and short-term heat treatments may induce diverse biophysical and physiological reactions in fruit of different cultivars. Additionally, it is evident that the mechanisms underlying the distinct cultivar-specific heat responsivity of apples are still poorly understood. Nevertheless, some of the cited and some of the above findings reported in this context may simply reflect differences in the stage of fruit development at optimal harvest ripeness of apples of the different cultivars.

### 4.3. Temperature Effects on Fruit Photosynthesis

Although sHWT in the time-frame of up to 60 s and the tested temperature range of up to 65 °C did not elicit any metabolic or structural damage, it nevertheless induced clear inhibition of photosynthetic performance beyond the temperature thresholds as indicated by CFI analyses. This further highlights the well-established suitability of this technique in the evaluation of fresh and minimally processed produce to pre and postharvest handling [12,13,16,17,26,42]. The presented results also indicate the high heat-stress responsiveness of the photosynthetic light energy captured by photosystem II. This high sensitivity can be instrumental in the understandings of the possible mechanism(s) underlying the rapid initial effects of sHWT on apples.

The range of threshold temperature obtained on apples of the different cultivars lies far beyond that normally reported as thermally stressing leaf photosynthesis of crop plants (e.g., T > 40 °C [12,42,43,44,45]). In these cases, leaves were exposed to heat for a longer time and reported temperature may be seen as mean steady state conditions. Serious heat-stress symptoms also occur in massive apples if heat-treated above 40 °C under steady-state conditions for prolonged time [39]. However, whole fruit steady-state temperatures are certainly not the case in short-term hot-water treated apples [2]. Heat-pulses treatments are generally known to yield higher temperature thresholds [18]. Nevertheless, there is generally a narrow gap between the inhibition of photosynthetic activity and damage of at least important parts of this complex metabolic machinery [12].

Concerning the general heat response of the overall photosynthetic apparatus, multiple sites for heat-induced impairment within the chloroplast membranes have been identified [46,47]. Especially PSII is more heat sensitive than the other components of photosynthetic electron transport chain, e.g., PSI [46], and energy generation. On the other hand, the enzymes of the photosynthetic Calvin cycle are assumed to be relatively thermostable [18,43,46,47,48]. Although the heat-dependent reduction of ribulose-1,5-bisphosphate carboxylase/oxygenase (Rubisco) activity due to the impaired Rubisco activase activity (e.g., [18] and literature cited therein) is assumed as a primary determinant of the reduction of photosynthetic performance [18], this response is already observed at moderate high temperatures (35 to 40 °C) and its reaction half-time is much longer than the duration of sHWT [46].

Nevertheless, short-term high temperature stress may rapidly induce phase transitions of the lipid double-layer of biomembranes, i.e., alters the permeability of the thylakoids, and thus reduces the transmembrane proton gradient [45], denaturates membrane-bound and solved proteins [49,50], changes the chloroplast organization, and affects the ultrastructure of the photosynthetic systems [47]. Furthermore, severe heat-stress is assumed to directly reduce the photosynthetic electron transport activity by disorganization of photosystem II (PSII) and of tertiary antenna structure and by destructing water splitting, and thus oxygen evolution [18]. Thus, heat damage as indicated by the rapid and irreversible decline of F_v_/F_m_ may involve effects on both proteins, e.g., major components of the PSII core complex [9] and on membranes [50]. This, however, does not necessarily include destruction of chlorophylls [42].

In the present study, the analyses of the respective measured fluorescence parameters may indeed allow a partial evaluation of the relevant mechanisms of photosynthetic impairment. In fact, only F_m_ declined beyond the critical temperature, while no variations in mean F_0_ were obvious. The decline in F_m_ may be indicative of an initial and rapid detachment of the light harvesting complexes of PSII (LHCII) from the photosystem cores, which reduces the absorption cross-section and thus the probability of the chlorophyll a fluorescence emission [45]. Similarly, Hüve et al. [18] found F_m_ to be more sensitive also to steady-state heat application than F_0_. Furthermore, in leaves of *Welwitschia mirabilis* heat stress reduced F_m_, while the dark adapted F_0_ slightly increased with heat stress [14].

On the other hand, the rise in the latter fluorescence parameter with continuously increasing temperatures at low rates is often used to evaluate the so-called critical temperature (T_c_ ) [15,51,52], above which inhibition of photosynthesis may increasingly become irreversible [18]. Then, membrane damage and oxidative stress may become serious, finally leading to visible lesions as described earlier [2,11].

As indicated by the constancy of F_0_ in fruit of all investigated cultivars, this critical temperature had quite obviously not yet been reached in the present study. Nevertheless, the factors responsible for the rise in F_0_ can also be partly reversible after transfer to lower temperatures [18]. For example, the oxygen evolving complex released from the PSII core by the heat action may rebind and the permeability of thylakoid membranes may decrease again. Although the initial chlorophyll fluorescence measurements were done shortly after the sHWT, the duration of the lack time until all fruit were treated may have been enough to allow for this partial recovery. Consequently, the effective critical temperature (T_Crit_) in this study must be defined as the critical temperature beyond which F_v_/F_m_ declines. In ‘Braeburn’ apples, F_v_/F_m_ only declined after exposure to 60 °C and did not recover during cold-storage. This all may indicate that the observed decline in F_m_ and, thus, F_v_/F_m_, resulted from some irreversible damages to the photosynthetic apparatus, even after the short-term temperature treatments used.

In contrast to ‘Braeburn’ apples, F_v_/F_m_ only declined beyond a threshold of 65 °C in fruit of the other cultivars. As discussed above, the ‘Braeburn’ apples seem to be very sensitive to sHWT. Similar to the present findings, Wand et al. [53] reported that ‘Braeburn’ apples freshly picked from the tree were more sensitive to long-term heat stress than those of ‘Fuji’ or ‘Cripps’ Pink’.

On the other hand, it was shown that photoprotection of light exposed apples of different cultivars during heat stress seem to pronouncedly vary in the respective protection mechanisms [22]. The photosynthesis of ‘Braeburn’ but also of ‘Granny Smith’ fruit appeared to depend more on the xanthophyll cycle for photoprotection than those of ‘Fuji’, ‘Golden Delicious’, and ‘Topred’. Delgado-Pelayo et al. [35] indeed showed that peels of dark stored apples of ‘Granny Smith’ contain five-times more xanthophyll cycle pigments than ‘Fuji’ and ‘Golden Delicious’ fruit. Interestingly, Li et al. [7] reported that the antioxidant capacity of yellow-greenish ‘Golden Delicious’ fruit was less impacted by long-term heat treatments at 60 °C, i.e., heat stress according to the temperature threshold in this study, than those of ‘Red Fuji’.

In general, presented results on the photosynthetic responses of apples treated at different temperatures for various durations suggest that serious damages at the cellular level may become relevant when a distinct critical heat dose was exceeded [18]. Hüve et al. [18] assumed that long-term (steady-state) heat treatments irreversibly impact the photosynthetic machinery at lower temperatures than rapid or dynamic heating, i.e., the degree of inhibition and finally damage depends on both the “heat pulse” temperature and on the time the fruit is exposed to this temperature [52]. On the other hand, presented results of the responses of ‘Braeburn’ apples imply that the impact of the factor time seems pronounced only below T_crit_ and less so beyond this temperature. Even at 55 °C, a 120 s-treatment resulted in less reduction of F_v_/F_m_ than a 30 s-treatment at 70 °C, which yielded the lower temperature dose (c.f. Figure 3). In addition, at temperatures beyond T_crit_, the damaging effect of HWT is also indicated by the lack of any recovery effect.

## 5. Conclusions

The finding that short-term hot-water-treatments are suitable for gentle decontamination of apples [11] was confirmed.

For evaluation of the optimal HWT processing parameters, chlorophyll fluorescence imaging (CFI) provided a rapid, sensitive, and effective approach. Variations in F_m_ and the maximum photochemical efficiency (F_v_/F_m_) closely reflected effects of sHWT on produce photosynthesis.

As indicated by the constancy of both NDVI and NAI, sHWT did not affect chlorophyll or anthocyanin contents of apples.

The responses of photosynthetic performance to sHWT conditions seemed to be cultivar-specific as were the optimal sHWT processing parameters in general.

Consequently, sHWT up to 55 °C for up to 60 s is gently and safely for apples under (semi-) practical conditions and may be applied without any impacts on fruit quality.

## Figures and Tables

**Figure 1 foods-09-00820-f001:**
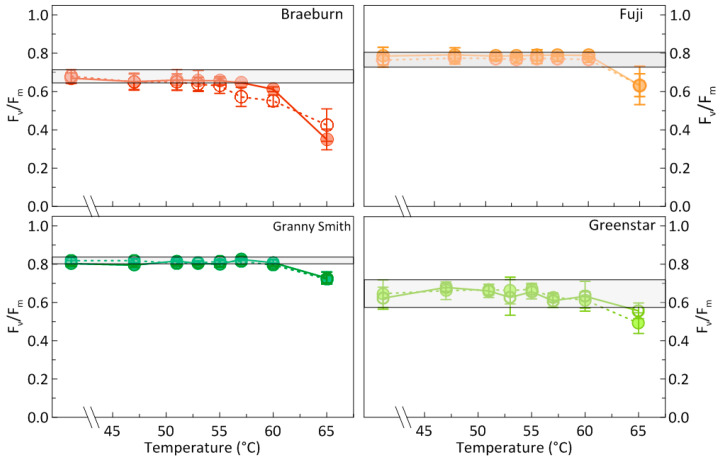
Means of the potential photon yield of photosystem II (F_v/_F_m_) of ‘Braeburn’, ‘Fuji’, ‘Granny Smith’, and ‘Greenstar’ apples, hot-water-treated at temperatures ranging from 46 to 65 °C for 30 s and measured one day (open circles, dotted lines) and eight days (filled circles) after the treatments. The light-grey horizontal bars indicate the standard deviation of results of the respective controls on day one of the experiment, roughly reflecting initial mean variability of F_v/_F_m._

**Figure 2 foods-09-00820-f002:**
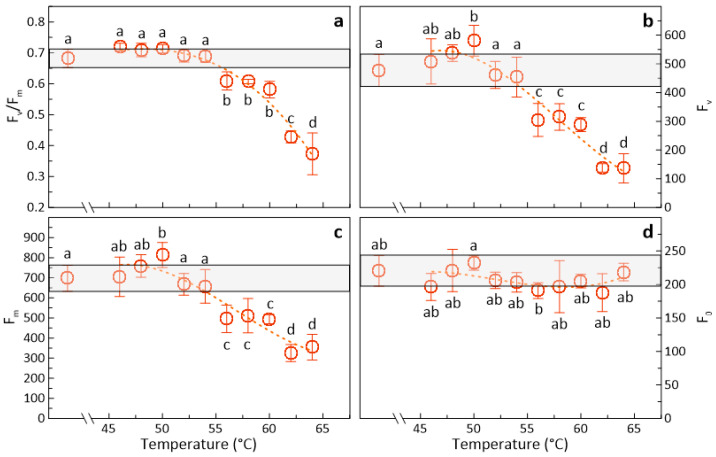
Average results of (**a**) the potential maximum photon yield of PSII (F_v/_F_m_), (**b**) the variable fluorescence (F_v_), (**c**) the maximum (F_m_), and (**d**) the dark fluorescence signal (F_0_) of ‘Braeburn’ apples (± SD; n = 3), hot-water-treated for 60 s at temperatures ranging from 46 to 64 °C. The light-grey horizontal bars indicate the standard deviation of results of the respective controls on day one of the experiment, while the dotted lines indicate respective polynomial fits. Different superscripts indicate significant differences between means (Duncan’s test, *p* < 0.05).

**Figure 3 foods-09-00820-f003:**
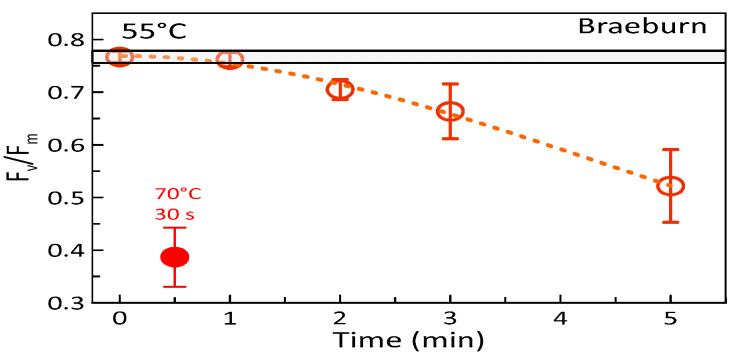
Means of the potential maximum photon yield of PSII (F_v/_F_m_) of ‘Braeburn’ apples (± SD; n = 3), hot-water-treated at 55 °C for 0, 60, 120, 180, and 300 s and the effect of hot water treatments at 70 °C for 30 s is also shown. The light-grey horizontal bars indicate the standard deviation of results of the respective controls on day one of the experiment, while the dotted line indicates a polynomial fit.

**Figure 4 foods-09-00820-f004:**
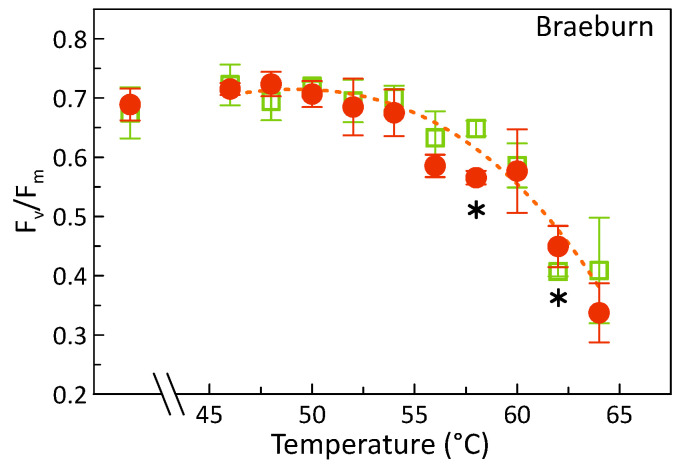
Means of the potential maximum photon yield of PSII (F_v/_F_m_), pixel-averaged for each of the two sides (mostly red: filled circles; mostly greenish: open squares) of the fruit separately analyzed per ‘Braeburn’ sample (± SD, see bars; n = 3). The asterisks denote significant (Duncan’s test, p < 0.05) differences between means.

**Figure 5 foods-09-00820-f005:**
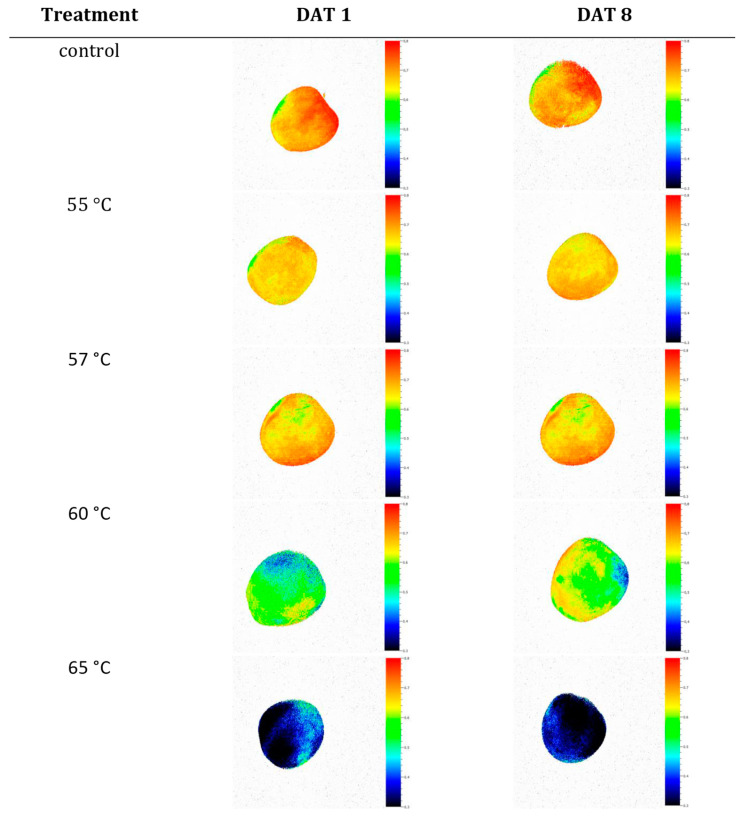
Local variations of the potential maximum photon yield of PSII (F_v/_F_m_) of ‘Braeburn’ apples, hot-water-treated at temperatures ranging from 55 to 65 °C for 30 s. Measurements were performed 1 or 8 d after treatments (DAT). Controls were not treated. After HWT and between measurements, all samples were stored at 4 °C.

**Figure 6 foods-09-00820-f006:**
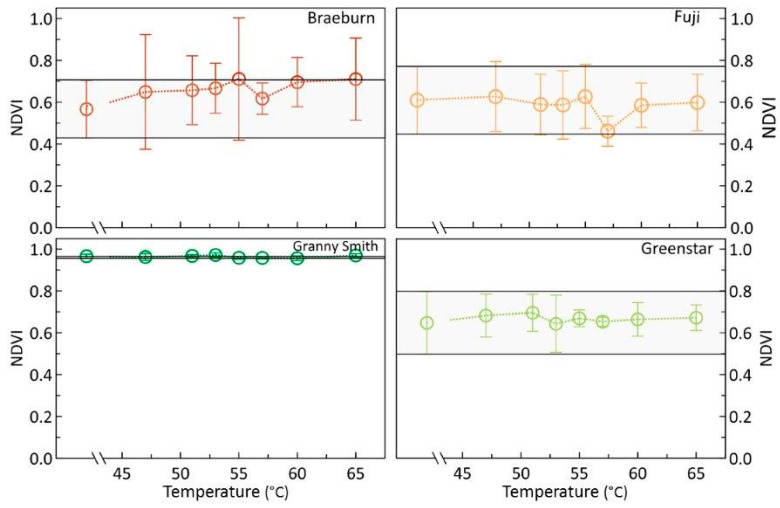
Means of the Normalized Difference Vegetation Index (NDVI) of ‘Braeburn’, ‘Fuji’, ‘Granny Smith’, and ‘Greenstar’ apples, hot-water-treated in the temperature range of 46 to 65 °C for 30 s. The light-grey horizontal bars indicate the standard deviation of results of the respective controls on day one of the experiment, roughly reflecting initial mean parameter variability.

**Figure 7 foods-09-00820-f007:**
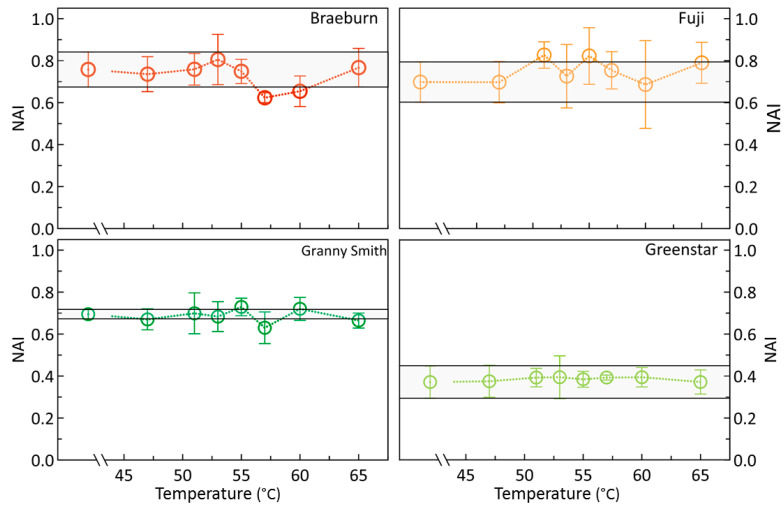
Means of the Normalized Anthocyanin Index (NAI) of ‘Braeburn’, ‘Fuji’, Granny Smith’, and ‘Greenstar’ apples, hot-water-treated in the temperature range of 46 to 65 °C for 30 s. The light-grey horizontal bar indicates the standard deviation of results of the respective controls, roughly reflecting initial mean parameter variability.

**Table 1 foods-09-00820-t001:** Means of vitamin C, total soluble solids (TSS), and titratable acidity contents (given as malic acid content in kg m^−3^), and tissue strength and stiffness (Young’s modulus) in untreated controls, and in fruit stored at 4 °C for 1 and 8 d after treatments (DAT). As HWT showed no effect on either parameter, values were averaged over all treated and untreated samples (± SD) to indicate differences between fruit of the different cultivars. Small letters indicate significance of differences between means.

	Vitamin C (g m^−3^)	TSS (%)	Malic acid (kg m^−3^)	Strength (N)	Stiffness (kN m^−1^)
‘Braeburn’ control	106.0 ± 21.6^a^	11.70 ± 1.10^a^	3.53 ± 0.80^a^	67.9 ± 2.9^a^	34.0 ± 3.6^a^
‘Braeburn’ DAT1	91.9 ± 18.2^ab^	11.70 ± 0.68^a^	3.80 ± 0.91^a^	55.5 ± 7.3^b^	28.3 ± 3.0^b^
‘Braeburn’ DAT8	83.6 ± 18.4^b^	11.70 ± 0.67^a^	3.78 ± 0.85^a^	48.5 ± 6.5^c^	25.4 ± 3.3^c^
‘Fuji’ control	76.2 ± 14.79^a^	15.38 ± 1.27^ab^	2.90 ± 0.28^a^	80.6 ± 7.5^a^	35.7 ± 0.3^a^
‘Fuji’ DAT1	65.7 ± 11.6^a^	14.27 ± 2.58^a^	3.06 ± 0.48^a^	79.2 ± 9.0^a^	33.3 ± 2.4^a^
‘Fuji’ DAT8	70.2 ± 11.9^a^	15.72 ± 2.51^b^	3.11 ± 0.56^a^	80.9 ± 6.2^a^	31.3 ± 4.0^b^
‘Granny Smith’ control	93.1 ± 5.4^a^	11.22 ± 0.24^ab^	9.70 ± 0.89^a^	98.5 ± 6.6^a^	39.9 ± 5.5^a^
‘Granny Smith’ DAT1	90.9 ± 14.7^a^	11.13 ±0.46^a^	10.55 ± 1.52^a^	96.6 ± 5.3^a^	39.9 ± 3.9^a^
‘Granny Smith’ DAT8	90.6 ± 14.7^a^	11.62 ±0.47^b^	10.08 ± 1.32^a^	92.7 ± 5.3^b^	38.8 ± 3.9^a^
‘Greenstar’ control	137.3 ± 10.8^a^	11.27 ±0.92^ab^	4.47 ± 0.99^a^	92.0 ±.9.6^a^	34.0 ± 5.7^a^
‘Greenstar’ DAT1	144.3 ± 14.2^a^	11.58 ±0.45^b^	4.51 ± 0.93^a^	94.0 ± 5.0^a^	34.2 ± 3.1^a^
‘Greenstar’ DAT8	135.1 ± 13.6^a^	11.12 ±0.48^a^	4.48 ± 0.83^a^	90.9 ± 5.8^a^	32.5 ± 2.9^a^

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
