# Peer review of "Optimization of Short-Term Hot-Water Treatment of Apples for Fruit Salad Production by Non-Invasive Chlorophyll-Fluorescence Imaging"

_foods, 2020, doi:10.3390/foods9060820_

Round 1

Reviewer 1 Report

Comments and Suggestions for Authors

The paper entitled “Optimization of short-term hot-water treatment of apples for fruit salad production by non-invasive chlorophyll-fluorescence imaging”. This work is really good and interesting.

Author Response

Dear reviewer,

thank you very much for your very positive comments.

Reviewer 2 Report

In the manuscript foods-778828 author studied the effect of short-term hot-water treatment of 3 apples for fruit salad production by non-invasive 4 chlorophyll-fluorescence imaging. Each of method has been explained properly in the study. However, some important information is missing without those it is incomplete study 1) Authors did not calculate the peel color parameters of short-term hot-water treated and untreated apple slices stored at 4 ◦C 2) Authors didn’t mention anything about the microbiological analysis 3) Author should provide pictorial data for the fresh and HWT apples. 4) Author has published recent studies on hot water treatment and the current study is unique from the previous.

Author Response

Dear reviewer,

thank you very much for your nice comments. To complete the report on our study, we would like to provide the important information that you were missing in the manuscript.

1) Authors did not calculate the peel color parameters of short-term hot-water treated and untreated apple slices stored at 4 â—¦C

For the presented study, we never intended to cut the apples and hence, could not analyse the surface colour of the fruit slices. In fact, such information has yet been published in detail in another paper that specifically dealt with this topic (Rux et al., 2019. Effects of pre-processing hot-water treatment on quality and shelf life of fresh cut apple slice. Foods 8, 653. https://doi.org/10.3390/foods8120653).

As stated in Abstract and Introduction, we concentrated on the systematic optimisation of temperatures and duration of the short-term hot-water treatment for apples, designated for fruit salad production, by non-invasive chlorophyll-fluorescence imaging.

2) Authors didn’t mention anything about the microbiological analysis

As explained in the Introduction section of the presented study, we never wanted to analyse the microbiological loads of either intact fruit or fresh cut slices. In fact, such information has yet been published in detail in two other papers, specifically dealing with these topics (Kabelitz & Hassenberg, 2018. LWT - Food Sci. Technol., 98, 492–499. https://doi.org/10.1016/j.lwt.2018.08.062 and Rux et al., 2019. Foods 8, 653. https://doi.org/10.3390/foods8120653).

3) Author should provide pictorial data for the fresh and HWT apples.

We prefer not to present additional RGB-images of treated apples because we had included chlorophyll fluorescence images, which provide much more information. In addition, such photos have yet been published in both Kabelitz & Hassenberg K, 2018, and Kabelitz et al. 2019. LWT - Food Science and Technology 108, 416–420. https://doi.org/10.1016/j.lwt.2019.03.067

4) Author has published recent studies on hot water treatment and the current study is unique from the previous.

Relevant results of this project (please see funding information in the manuscript) have been published in three other papers (Kabelitz & Hassenberg, 2018; Kabelitz et al., 2019 and Rux et al. (2019), as cited in References of this manuscript. Each of these papers dealt with a specific aspect of the entire project and no results were used twice.

Reviewer 3 Report

General comments: In Herppich et al., the authors intended to use Chlorophyll a fluorescence imaging for measuring effects of HWT on four types of apple fruits. From the abstract, initially, it seemed that the objective was to measure the effects of HWT to protect fruits from microbial spoilage and human pathogens. In fact, I was trying to find an answer to an obvious question that how many minutes of HWT is required to eliminate, if any, microbial spoilage or pathogens from fruits, but it turned out to be an interesting use of ChlF imaging for measuring health of apple fruits. They indeed measured Fv/Fm and other parameters of the apple fruits post HWT at different temperatures and also measured different timing of HWT probably in most sensitive apple fruit. It seems authors have collected more data that was not effectively presented in the manuscript; moreover, they have given a vague analysis of the data. 

I advise authors to read, for example, Photosynthesis Research. 139(1-3): 123-143, and other related articles to be better informed on the use of ChlF imaging. The authors may re-submit or the manuscript after a major revision of all parts of the manuscript to this journal. 

Important notes to improve manuscript is following. Authors need to carefully improve the

Abstract: Authors should write “Chlorophyll a fluorescence” throughout the manuscript instead of” chlorophyll fluorescence”. Also, the abbreviation of “Chlorophyll a fluorescence” should be “ChlF”. For example, “chlorophyll a fluorescence imaging “should be abbreviated as ChlFI.

#L#26

 “remission spectroscopy“? Usually, it is well known as reflectance spectroscopy.

#L#30. One cultivar was sensitive at 55 °C, but if I understood well, Fv/Fm of the other three cultivars was sensitive at 60 °C. Is not it? Again, here my question is valid as it is not answered HWT at what temperature is good enough for eliminating microbes and pathogens.

I would suggest to re-write the abstract with more emphasis on actual results.

Introduction: Introduction is very short and misleading because the Authors escaped citing many new important pieces of literature. I would write a very short paragraph or maybe within an existing paragraph on the use of ChlF imaging in different applied fields (see a review, Indian Journal of Plant Physiology. 21(4), 514-527; and citations used therein). Further authors should use relevant citations to properly write and define the parameter, Fv/Fm, and others used as per standard literature.   

M & Methods:

It was not clear how the authors measured Fv/Fm? Usually, in leaves we dark-adapt plant or leaves for about 20-30min, to oxidize QA, before the measurement of Fv/Fm.

Results:

Line 131-134: Fv/Fm measures “Maximum quantum yield of PSII photochemistry” not “photosynthetic capacity” as the author mentioned. From their text, it seems that they gave HWT at different times, e.g., 30 s and more, but time variation is not presented in the graph. Therefore, it would be more informative for readers to insert data on different times of HWT.

Line 139-141: Authors wrote that closer inspection showed that there is an improvement in Fv/Fm values post 1 week of storage, but they did not present that data. I think this is very important to put this data in Figure 2 if possible. Or maybe they can present those data in a Table if there is an overlap.

Line 144-148:

1) “In these……..Figure1”; needs revision as it is not clear 60 s at which temperature?

2) The authors should use second instead of min as a unit.  

3) If the difference between Fv/Fm measured for control and after 60 s HWT is insignificant, the author must avoid writing unscientific and vague words like “somewhat lower”

 149: Variation in Fv/Fm is only because of changes in Fm because Fo seems constant. Fv=Fm-Fo; therefore differences in variable fluorescence are indeed due to Fm.

153: It is not clear what authors meant as additional variability?

176: Does it mean that HWT after one week of storage at 4 degrees results in a similar pattern of Fv/Fm? The authors must present the data. The authors can’t summarize all results without presenting data. They must provide data that may be as additional supplementary files.

177-178: How authors justify a statement like “it also proved that the HW-treatment technique used indeed allows for uniform heating of the complete fruit surface “

Figure 1: I wonder why authors used “horizontal lines “which they named as “Horizontal bar” in the legends. 

This presentation seems very unusual. I advise them to use a simple “error bar” on both sides of averaged mean, in each data set, to represent the error. What is a dotted line along with the data sets? It seems the author plot fits, but what is the significance of plotting fits in this very simple data?

Figure 2: For “Horizontal bar” I stand my opinion similar to in Figure 1. This presentation seems very unusual. I advise them to use a simple “error bar” on both sides of averaged mean, in each data set, to represent the error. What is the significance of plotting fits in this simple data? it seems that authors did some t-test and calculated significance level; however, details have not been provided in the legends.

Figure 3: Comments in Fig 1 and 2 are applicable here too.

Figure 4: Legend is vague and not self-explanatory. It is not clear what do they meant by “each of the two sides”. In the same Figure, they wrote as shown in Figure 4. Authors must avoid phrases like “more or less complete or incomplete”. Again, the Authors discuss legends of Figure 4 to be explained in Figure 5. It seems

Figure 5: not “d after treatment” but “days after treatment”. They wrote, 1 or 8 DAT. Does it mean that following HWT, they did measurement the next days and then after 8 days and they probably transferred apples at 4 degrees immediately after the HWT? Clarify, please.

Figure 6: Follow earlier comments to improve the contents of this Figure. Combine data of all apple types in one Figure.

Figure 7: Combine as suggested in Figure 6 and comments for improving this Figure is valid.

I advise authors to combine Figure 6-7.  

Author Response

Dear reviewer,

thank you very much for your nice comments. To complete the report on our study, we would like to provide the important information that you were missing in the manuscript. We could follow nearly all your helpful recommendations. For the few cases that we could not, we carefully explained the respective reasons for the decision. We really hope that you can agree with our opinion.

Please find a point to point response to the comments, the raised problems and questions.

# the authors intended to use Chlorophyll a fluorescence imaging for measuring effects of HWT on four types of apple fruits.

As has been successfully done before in many other studies (please see https://www.researchgate.net/profile/Werner_Herppich for closer details), we used CFI to simply and rapidly evaluate the effects and to optimise the application of hot water treatments on fruit designated for fresh-cut salad production of four practically used apple cultivars in this study.

From the abstract, initially, it seemed that the objective was to measure the effects of HWT to protect fruits from microbial spoilage and human pathogens. In fact, I was trying to find an answer to an obvious question that how many minutes of HWT is required to eliminate, if any, microbial spoilage or pathogens from fruits, but it turned out to be an interesting use of ChlF imaging for measuring health of apple fruits.

As usual in abstracts, the scientific and the practical back ground of the current investigation were stated at the very beginning in the initial sentences. However, we realised that this probably was misleading for people not familiar with this topics of food science. Therefore, we now shortened the initial part and reduced the text to its very minimum. Nevertheless, we are pleased to read that you agree that CFI is a very helpful tool to test the performance and the sustainability of postharvest treatments. This is an approach that we successfully use since nearly two decades.

They indeed measured Fv/Fm and other parameters of the apple fruits post HWT at different temperatures and also measured different timing of HWT probably in most sensitive apple fruit.

Yes, you are correct; we presented all data necessary to rapidly and sensitively evaluate the temperature effects on intact apples, designated for fresh-cut salad production, under semi-practical conditions.

It seems authors have collected more data that was not effectively presented in the manuscript;

We presented all data necessary to rapidly and sensitively evaluate the effects of temperature and duration of the short-term hot-water treatment on intact apples under semi-practical conditions.

moreover, they have given a vague analysis of the data.

You may be sure that the analyses are competent and by no means vague.

I advise authors to read, for example, Photosynthesis Research. 139(1-3): 123-143, and other related articles to be better informed on the use of ChlF imaging.

We apologise that we did not cite your article. However, the recommended paper deals with low temperature induced modulations of photosynthetic induction in Arabidopsis thaliana and, as such, it does not touch the topic of the presented manuscript. In addition, despite your opinion, we are quite familiar with both CFA and with CFI. For nearly 35 years, one of us is applying CFA in basic and, especially, applied research; e.g. see von Willert, Matyssek and Herppich (1995) Experimentelle Pflanzenökologie: Grundlagen und Anwendungen. Georg Thieme Verlag, Stuttgart. ISBN 3-13-134401-6; or Herppich WB (2002) Application potential of chlorophyll fluorescence imaging analysis in horticultural research. In: Fruit, Nut and Vegetable Production Engineering, 609-614; and literature cited therein and in this manuscript. For more details, please see https://www.researchgate.net/profile/Werner_Herppich

Important notes to improve manuscript is following. Authors need to carefully improve the 

Abstract: Authors should write “Chlorophyll a fluorescence” throughout the manuscript instead of” chlorophyll fluorescence”. Also, the abbreviation of “Chlorophyll a fluorescence” should be “ChlF”. For example, “chlorophyll a fluorescence imaging “should be abbreviated as ChlFI.

No general rule exists, how to abbreviate chlorophyll fluorescence, chlorophyll fluorescence analysis or chlorophyll fluorescence imaging. As we did not deal with the basics of CF processes but used CFI for optimisation of a practical postharvest treatment, it is also not relevant for this purpose that, for kinetical reasons, the measured fluorescence signals evolve from chl a molecules of PSII (for more detail, please c.f. von Willert et al. (1995) or Matyssek & Herppich, 2018, Chlorophyllfluoreszenzanalyse. In: Experimentelle Pflanzenökologie. Springer Reference Naturwissenschaften. Springer Spektrum, Berlin, Heidelberg, pp. 1-56. https://doi.org/10.1007/978-3-662-53493-9_13-1 ).

#L#26 “remission spectroscopy“? Usually, it is well known as reflectance spectroscopy.

Simply speaking, the term “spectroscopy” normally comprises the measurement of the basic optical properties, reflectance, absorbance and transmittance. For proper measurements of reflectance generally a so-called Ulbricht sphere-based instrument is used. If, however, some radiation penetrates the object, in our case the fruit, the photons can interact with e.g. pigments and are either absorbed and then partially re-emitted (therefore remission) again, or scattered and then re-emitted again or partially transmitted. The instrument used in this study is optimised for remission measurements. This approach provides closer information on the internal properties than normal reflection spectroscopy. For more detail see papers of M. Zude or B. Herold, e.g. in: Optical Monitoring of Fresh and Processed Agricultural Crops, 2008, M. Zude (Ed.), CRC Press; or see Kuckenberg et al., 2008; Postharvest Biology and Technology 48(2):231-241.

#L#30. One cultivar was sensitive at 55 °C, but if I understood well, Fv/Fm of the other three cultivars was sensitive at 60 °C. Is not it? Again, here my question is valid as it is not answered HWT at what temperature is good enough for eliminating microbes and pathogens.

I would suggest to re-write the abstract with more emphasis on actual results.

As stated in the Abstract and the Introduction, the objective of this manuscript was the rapid and simple cultivar-specific optimisation of HWT in terms of maximal processing temperature and duration without harmful effects on the produce. We did not state that this manuscript aims to improve or adapt the sanitation efficiency (see Kabelitz & Hassenberg, 2018). So, everything relevant is included in this Abstract, which is clearly and easily to read and we see no need for further changes.

However, as we had realised that the original initial sentences might probably be somewhat misleading for people not familiar with this topics of food science, we now shortened the initial part and reduced the text to its very minimum.

Introduction: Introduction is very short and misleading because the Authors escaped citing many new important pieces of literature.

We apologise that we did not cite your article. However, sufficient original and primary literature, necessary to understand the scientific background of this study is included. We apologise, but we see no need to also include Indian Journal of Plant Physiology. 21(4), 514-527.

I would write a very short paragraph or maybe within an existing paragraph on the use of ChlF imaging in different applied fields (see a review, Indian Journal of Plant Physiology. 21(4), 514-527; and citations used therein). Further authors should use relevant citations to properly write and define the parameter, Fv/Fm, and others used as per standard literature.

Thank you very much for this advice. We do not doubt that your review is a valuable contribution to literature, and we greatly apologise for not having cited it. However, we have yet included mostly original, primary literature, necessary to understand the scientific background of this study. We also felt it more helpful to relate to own relevant publication on the use of CFI particular in different relevant applied fields.

M & Methods:

It was not clear how the authors measured Fv/Fm? Usually, in leaves we dark-adapt plant or leaves for about 20-30min, to oxidize QA, before the measurement of Fv/Fm.

We feel that the measurement of Fv/Fm has been adequately described in the MaM section. CFA is meanwhile widespread applied (probably too wide spread?) that providing closer details is superfluous. On the other hand, we are aware that the recommendation to use 20 to 30 min or more for dark-acclimation is stated in many papers. The closer analyses of the dark relaxation kinetics, however, showed that the “optimal” “dark adaptation” time largely dependents on the specific measuring conditions. If there is no risk of photoinhibition, 2 to 3 min is enough to fully relax qE and another 7 to 10 min to fully eliminate any qT. We assume that the reviewer knows the respective relevant literature. So, as a compromise, 15 min are sufficient under nearly all conditions. In addition, during this time span also Rubisco and the “dark reaction” are completely deactivated. On the other hand, in cases of photoinhibition, the time needed to relax all components of non-photochemical quenching is very difficult to assess, simply because qI may need several hours to relax in severe cases. Consequently, 15 min (i.e. covering qE and, potentially, qT but also Rubisco etc.) is a highly recommended compromise. Any longer time may, in fact, introduce uncertainties due to qI (see Matyssek and Herppich, 2018, Chlorophyllfluoreszenzanalyse. In: Experimentelle Pflanzenökologie, Springer Reference Naturwissenschaften, https://doi.org/10.1007/978-3-662-53493-9_13-1; or, more specifically, Herppich et al., 1997;,Flora 192: 165-174. https://doi.org/10.1016/S0367-2530(17)30773-9; or Herppich et al., 1998, Physiologia Plantarum 102:148-154. https://doi.org/10.1034/j.1399-3054.1998.1020119.x).

Results:

Line 131-134: Fv/Fm measures “Maximum quantum yield of PSII photochemistry” not “photosynthetic capacity” as the author mentioned.

We fully agree to reviewer that Fv/Fm is a good “measure” of the “potential maximum quantum yield of electron flow through PSII“ (or similar) and, as such, indeed “reflects” the potential “photosynthetic capacity”. Many years ago, Björkman and Demmig (1987) showed the close relationship between Fv/Fm, photosynthetic O2 production and total photosynthesis. Hence, Fv/Fm is also an indicator of the maximum photochemical efficiency of total photosynthesis, except in case of DCMU, which itself was originally used to induce Fm before the invention of the saturation pulse method. But for more and closer details the reviewer might consult Matyssek and Herppich, 2018.

From their text, it seems that they gave HWT at different times, e.g., 30 s and more, but time variation is not presented in the graph. Therefore, it would be more informative for readers to insert data on different times of HWT.

In the Abstracts, in the Introduction and in the Materials and Methods section, the design of the experiments is clearly stated. Furthermore, in the heading of each figure, this information is given again. In fact, it is certainly not helpful if not impossible to include all information of all the different experiments in just ONE figure. Therefore, we preferred to present the results of the different experiments in different figures. We are sure that this is also the notion of the reviewer.

Line 139-141: Authors wrote that closer inspection showed that there is an improvement in Fv/Fm values post 1 week of storage, but they did not present that data. I think this is very important to put this data in Figure 2 if possible. Or maybe they can present those data in a Table if there is an overlap.

Thank you very much for this comment. This was indeed not included in the heading of Figure 1. We now added this information to the heading.

Line 144-148:

1) “In these……..Figure1”; needs revision as it is not clear 60 s at which temperature?

We do not understand this comment. The text clearly and correctly stated that “analysed in more detail in fruit of this cultivar (Figure 2). In these experiments, treatments for 60 s did not further affect the overall temperature response of Fv/Fm as compared to those shown in Figure 1.

From normal English grammar, there is no doubt that “In these experiments” can only be related to “more detail in fruit of this cultivar (Figure 2).” Furthermore, the next part states that these results should be “compared to those shown in Figure 1.” Again, this is correct and easy to understand. So, there is no need to change the sentences.

2) The authors should use second instead of min as a unit.

We now consistently use “s”, although we do not feel that someone could be confused by using both “min” and “s”.

3) If the difference between Fv/Fm measured for control and after 60 s HWT is insignificant, the author must avoid writing unscientific and vague words like “somewhat lower”

Writing “somewhat lower” is certainly neither unscientific nor vague, but it is a mathematical fact that the means are indeed “somewhat lower”, completely independently for any statistic. In this context, I am pretty sure that the reviewer agrees that even the best statistical test in merely an estimation of probabilities and can’t prove of negate anything. Assuming something else is simply a misinterpretation of statistics, unfortunately widespread in literature.

 149: Variation in Fv/Fm is only because of changes in Fm because Fo seems constant. Fv=Fm-Fo; therefore differences in variable fluorescence are indeed due to Fm.

We fully agree. This is exactly what is written in this sentence. Even when only Fm changes, but F0 remains constant, then Fv changes as well because Fv = Fm – F0. As a consequence, Fv/Fm changes due to changes in the variable fluorescence signal (Fv) and in the maximum fluorescence signal (Fm).

153: It is not clear what authors meant as additional variability?

It is not clear for us, why this is not obvious. If, as proposed, Fig. 2a and b are compared, it shows that the variability of Fv is larger than that of Fv/Fm, without providing additional information relevant for the evaluation of the heat effects.

176: Does it mean that HWT after one week of storage at 4 degrees results in a similar pattern of Fv/Fm? The authors must present the data. The authors can’t summarize all results without presenting data. They must provide data that may be as additional supplementary files.

It is not clear for us what the reviewer really means. Instead, in line 170 one reads “This was also valid for all fruit after one week of storage (Figure 5).” This is formally correct because in Fig. 5, both results of day 1 and day 8 are presented. Irrespective of this, it is common use in scientific publication, not to show unnecessary results, but to shortly mention it as “data not shown” if it is only a repetition of results yet presented. So, again, it is not clear for us what the reviewer really means.

177-178: How authors justify a statement like “it also proved that the HW-treatment technique used indeed allows for uniform heating of the complete fruit surface “

From all results presented in line 161 to 173 it is clear that heating was indeed uniformly over the complete fruit surface. Hence, it is not clear for us what the reviewer means.

Figure 1: I wonder why authors used “horizontal lines “which they named as “Horizontal bar” in the legends. This presentation seems very unusual. I advise them to use a simple “error bar” on both sides of averaged mean, in each data set, to represent the error.

Using bars, either vertical or horizontal, is common in scientific literature. As can easily be seen, it is indeed a bar and not only “horizontal lines”. We feel that this presentation style facilitates reading and understanding of the figure und prefer to leave the figure as it is.

What is a dotted line along with the data sets? It seems the author plot fits, but what is the significance of plotting fits in this very simple data?

Again, it should simply help to follow the responses of the parameters and is not used statistically. Again, we feel that this presentation style facilitates reading of the figure und prefer to leave it as it is.

Figure 2: For “Horizontal bar” I stand my opinion similar to in Figure 1. This presentation seems very unusual. I advise them to use a simple “error bar” on both sides of averaged mean, in each data set, to represent the error. What is the significance of plotting fits in this simple data? it seems that authors did some t-test and calculated significance level; however, details have not been provided in the legends.

Figure 3: Comments in Fig 1 and 2 are applicable here too.

For “horizontal bar”, please see above. Even if the “presentation seems very unusual“ for the reviewer, in fact, it is not. We added some pieces of information on the exact meaning of the small letters to the figure headings. Closer information on statistics is, as usual, given in the Materials and Methods section (lines 126 ff).

Figure 4: Legend is vague and not self-explanatory. It is not clear what do they meant by “each of the two sides”.

To make the heading self-explanatory for everyone, we added the information “of the fruit”.

In the same Figure, they wrote as shown in Figure 4.

To be honest, we do not understand this comment.

Authors must avoid phrases like “more or less complete or incomplete”.

We feel that this is neither unscientific nor vague. In this specific context, it is related to a description of ‘Braeburn’ apples. As real biological objects, apples are never “ideal”. In particular, fruit of this cultivar never show a fully “red” or “green” or “yellowish” side but the respective appearance is indeed “more or less complete or incomplete”. This example also highlights the ineffectiveness of a simple statistical view on biological objects.

Again, the Authors discuss legends of Figure 4 to be explained in Figure 5.

To be honest, we do not understand this comment.

Figure 5: not “d after treatment” but “days after treatment”. They wrote, 1 or 8 DAT. Does it mean that following HWT, they did measurement the next days and then after 8 days and they probably transferred apples at 4 degrees immediately after the HWT? Clarify, please.

The full text is “1 or 8 d after treatments”, so a number is correctly followed by a unit symbol, as is urgently required by the refined SI rules:

  • SI requires that numerals be followed by proper SI units; e.g. 12 d, not 12 days. Likewise words should not be followed by unit abbreviations; e.g., twelve days, not twelve d.

In addition, the design of the experiment is clearly stated in the MaM section as is appropriate for a scientific manuscript. A figure heading should certainly be complete but it should also be short.

Figure 6: Follow earlier comments to improve the contents of this Figure. Combine data of all apple types in one Figure.

Figure 7: Combine as suggested in Figure 6 and comments for improving this Figure is valid.

I advise authors to combine Figure 6-7.

To be honest, we see no advantage it these advices. In contrast, this recommendation would make a combine figure too large and too complex, thus, hampering readability. Thus, we prefer to leave them as they are.

Round 2

Reviewer 2 Report

All the comments had been addressed in the revised manuscripts.